# Contemporary Molecular Analyses of Malignant Tumors for Precision Treatment and the Implication in Oral Squamous Cell Carcinoma

**DOI:** 10.3390/jpm12010012

**Published:** 2021-12-28

**Authors:** Julia Yu Fong Chang, Chih-Huang Tseng, Pei Hsuan Lu, Yi-Ping Wang

**Affiliations:** 1Graduate Institute of Clinical Dentistry, School of Dentistry, National Taiwan University, Taipei 10617, Taiwan; jyfchang@ntu.edu.tw (J.Y.F.C.); 1050565@kmuh.org.tw (C.-H.T.); r04422025@ntu.edu.tw (P.H.L.); 2Department of Dentistry, National Taiwan University Hospital, College of Medicine, National Taiwan University, Taipei 10617, Taiwan; 3Graduate Institute of Oral Biology, School of Dentistry, National Taiwan University, Taipei 10617, Taiwan; 4Division of Oral Pathology & Maxillofacial Radiology, Kaohsiung Medical University Hospital, Kaohsiung 80756, Taiwan; 5Division Oral & Maxillofacial Imaging Center, College of Dental Medicine, Kaohsiung Medical University, Kaohsiung 80708, Taiwan

**Keywords:** oral squamous cell carcinoma, personalized medicine, therapy

## Abstract

New molecular tests and methods, in addition to morphology-based diagnosis, are widely used as a new standard of care in many tumors. “One-size-fits-all medicine” is now shifting to precision medicine. This review is intended to discuss the key steps toward to development of precision medicine and its implication in oral squamous cell carcinoma. The challenges and opportunities of precision medicine in oral cancer will be sequentially discussed based on the four steps of precision medicine: identification/detection, diagnosis, treatment and monitoring.

## 1. Introduction

Head and neck cancers were the eighth-most common cancer worldwide in 2020, affecting 878,000 new patients and causing 444,000 deaths [1]. The incidence of oral cancer in Taiwan is 32.46 per 100,000 persons, which is the highest in the world [2]. Smoking and betel nut chewing are implicated in the rise of head and neck cancers in developing countries, but in contrast, the number of human papillomavirus (HPV)-associated oropharyngeal cancers, which is mainly induced by HPV type 16, is increasing in developed countries [3]. Due to different etiologies, the focus here is on oral squamous cell carcinoma (OSCC). The major etiologic factors for OSCC are cigarette smoking, alcohol drinking and betel nut chewing. The current standard treatment is surgical resection with radiation and chemotherapy as adjunct therapies. High risk of local recurrence and some distant metastasis are seen in locally advanced disease and are associated a poor prognosis with less than 50% 5-year overall survival. High-doses of platinums, taxanes and antifolates-based chemotherapy administered intravenously are the major drugs for systemic therapy. Cetuximab, pembrolizumab and nivolumab are the current targeted therapy drugs for OSCC [4,5,6]. Previously investigated molecular and clinical risk factors have shown limited clinical utility [7]. Precision medicine shows promising results in the treatment of several leading solid tumors, including lung [8] and breast cancers [9]. This article reviews the key steps, identification/detection, diagnosis, treatment and monitoring, toward the progress of precision medicine in these successful examples and the implication in OSCC (Figure 1).

Traditionally, malignant tumors are classified based on the cell of origin, differentiation, histopathogenesis and other morphological and immunohistochemical characteristics. However, the molecular heterogeneity in the same group of tumors, even in the same tumor is responsible for the different responses to the same treatments among the patients. As the classic histopathological diagnosis followed by a universal treatment can only partly control the disease and predict the prognosis. New molecular tests and methods in addition to morphology-based diagnosis are now widely used as a new standard of care in many tumors. “One-size-fits-all” medicine is now shifting to precision medicine (Figure 2). 

Although the definitions of precision and accuracy are distinct semantically, the US National Research Council included both concepts of precision and accuracy in its definition of precision medicine [10]. Moreover, precision medicine and personalized medicine are frequently used interchangeably. However, the US National Research Council distinguished these two terms and defined the personalized medicine as the therapies are synthesized for specific individuals [10], which is thought to be the final goal for precision medicine (Figure 2). From classic chemotherapy, targeted therapy to Chimeric Antigen Receptor T-cell (CAR-T) immune therapy, medicine is evolving toward precision and personalized medicine.

The early successful application of molecular classification and targeted therapy, which was the beginning of a new paradigm for precision oncology, is the development of tyrosine kinase inhibitor (TKI) imatinib to treat patients with chronic myeloid leukemia (CML), caused by constitutively active BCR-ABL1 fusion tyrosine kinase [11]. This example demonstrates the translation of basic science discoveries to clinical application. The first step is identification of the target, here the Philadelphia chromosome, later proved as a consistent translocation between chromosomes 9 and 22 that resulted in a BCL-ABL1 fusion. The second step is using the molecular methods for diagnosis, such as Fluorescence in situ hybridization (FISH) and polymerase chain reaction (PCR)-based genetic molecular testing or sequencing. The third step is applying the proper medication, imatinib, in this selected group of patients. The final step is monitoring the disease progress and recurrence [11]. Since then, cancers have been recognized as genetic diseases involving gene alterations of various cell functions, including but not limited to cell proliferation, differentiation, DNA repair, and apoptosis [12]. 

Compared to the hematologic diseases, solid tumors are more complex and involve not only tumor cells but also their microenvironments [13]. Thus, the application of precision medicine in solid tumors began later than in hematologic neoplasms. However, precision medicine is now a reality for patients with non-small cell lung cancers (NSCLC). Comparison of the most updated cancer treatment guideline from The National Comprehensive Cancer Network^®^ (NCCN^®^) for non-small cell lung cancers [14] and head and neck cancers [15] is shown in Figure 3. 

For lung cancer patients presenting with advanced or metastatic diseases clinically, a biopsy to diagnose the histologic types will be performed first. If the pathologic diagnosis belongs to NSCLC group with the histologic types either as adenocarcinoma, large cell, squamous cell carcinoma or not otherwise specified (NOS), then the patients’ cancer tissues will be sent for further molecular testing. Different molecular tests including immunohistochemistry, fluorescence in situ hybridization (FISH), PCR-based analysis or next generation sequencing (NGS) for different gene changes will be performed. Then the proper targeted therapy will be used according to the corresponding molecular alterations based on the results from clinical reports or trials. Patients will be monitored regularly during the treatment. Tumor resistance sometimes occurs after treatment. When tumor resistance or progression occurs, the next run of biomarker testing will be started again. The second run of the treatment and monitoring will follow. This is the current precision treatment for NSCLC patients. 

Compared to NSCLC, PD-L1 testing is the only currently available biomarker testing for patients with head and neck cancers in the most recent NCCN guidelines [15] (Figure 3). As precision medicine is more advanced in top leading cancers, including lung, breast, colorectal, and skin cancers, than in oral cancers, this review is intended to discuss the key steps toward development of precision medicine in these leading cancers and its implication in oral squamous cell carcinoma. The challenges and opportunities of precision medicine in oral cancer will be sequentially discussed based on the four steps of precision medicine, identification/detection of target, diagnostic tests, treatment and monitoring. 

## 2. Identification/Detection of the Target

### 2.1. Definition for Good Targets and Target Classes

The “druggable” target is defined as a protein, peptide or nucleic acid with activity that can be modulated by a drug [16]. Although various genetic alterations can be identified in tumors, the number of appropriate drug targets is still limited [16]. Only around 4% of drug development programs are ultimately approved as licensed drugs [17]. Thus, a key factor in precision medicine is finding a good drug target. A good drug can bind to and modulate a molecular target in a safe and effective way when administered for a disease. The properties of an ideal drug target include that the target has a key role in the pathophysiology of a disease process, modulation of the target can be effectively in a defined patient population with no or less important effect under physiological condition or in other diseases [18]. 

The current targeted drugs belong to small molecular weight chemical compounds (SMOL) and biologics (BIOL). The target classes hit by SMOL include enzymes, receptors, transcription factors, ion channels, transport proteins, protein–protein interface, nucleic acids and the target classes hit by BIOL include extracellular proteins, transmembrane receptors, cell surface receptors, substrates and metabolites and RNA [16]. The modes of action for these targets are inhibitors or activators for the enzymes and transcription factors; agonists, antagonists, modulators, allosteric activators or sensitizers for receptors; inhibitors or openers for ion channels; inhibitors for transport proteins and protein–protein interface; alkylation, complexation and intercalation for nucleic acids; antibodies for extracellular proteins; recombinant proteins for transmembrane receptors and extracellular proteins; antibody–drug conjugates for cell surface receptors, enzymatic cleavage for substrates and metabolites and RNA interference for RNA [16]. 

For lung cancer treatments, the NCCN guidelines are established based on the knowledge of the common molecular changes, which also have been known as targets, are different in small cell or NSCLC. This knowledge depends on a large amount of research to identify the links between targets and disease states. Once the targets have been identified, validation of the targets, initiation of drug discovery programs and conduction of clinical trials will be followed. About 5–55% of all patients with NSCLC harbor mutations in *EGFR*, 8–30% with *KRAS* mutations, 3–5% with *ALK* fusion, 2–3% *HER2* mutation, 1–2% *RET* fusion, 1–3% *ROS1* fusion and 0.5–3% *BRAF* mutation based on the data in different ethnicity [19]. These genes are all driver oncogenes and currently have corresponding targeted therapies for these gene changes [9]. The targeted therapies for *EGFR*, *KRAS*, *ALK*, *MET*, *BRAF*, *RET*, *ROS1*, *NTRK* and *HER2* mutations are belong to SMOL inhibitors and the targeted therapies for *HER2* amplifications belongs to BIOL antibody drug conjugates [8]. So far, driver oncogenes show more promising results as targets for developing targeted therapies. 

### 2.2. Target Assessment

To facilitate the transition from identifying new drug targets, understanding the target biology, linking targets and disease states to testing drug candidates in clinical trials, the German Federal Ministry of Education and Research (BMBF) funded the GOT-IT (Guidelines on Target Assessment for Innovative Therapeutics) working group to establish a structured framework for target assessment. Five assessment blocks are suggested to characterize the targets: target–disease linkage (the causal relationship between target and disease), safety aspects (on target or target related), microbial targets (related to non-human targets), strategic issues (clinical needs and commercial potential) and technical feasibility (including drugability, assayability and biomarker availability) [18]. Traditionally, the target–disease linkage and safety aspects might be the most important issues in academic drug discovery. However, commercial potential and technical feasibility can also be crucial in viewpoints of pharmaceutical companies. If the target is a non-human target, such as a microbial target, then the third assessment blocks will be applied. Thus, these five assessment blocks can run in parallel or rearranged based on the project goals. 

Using lung cancer as an example, lung cancer is one of the leading causes of cancer death, making up almost 25% of all cancer deaths [8]. This epidemiological data fits the assessment block of strategic issues (clinical needs and commercial potential). The *EGFR* mutation is the most common identified gene alteration among lung cancer patients after a large amount research using a variety of methods, such as in vitro cell culture, ex vivo, in vivo animal models and human samples. The causal relationship between *EGFR* mutation and lung cancer is then established. After technical feasibility assessment, such as drugability assessment, safety assessment and biomarkers availability assessment, is performed, then treatment of lung cancers by tyrosine kinase inhibitors of EGFR can be proved and further evaluated in clinical trials and finally as global treatment guidelines. 

### 2.3. Challeges and Future Directions in Oral Cancer

Since there are some similar features between lung and oral cancers, such as the major risk factor being smoking, that both patient groups can be broadly divided into patients with risk factors and without risk factors and the major genetic change being a *TP53* mutation, we mainly used the development of precision medicine in NSCLC to be a role model for developing precision medicine in oral cancer here.

From the precision medicine applied in NSCLC patients, it is worth noticing that most lung cancer patients who harbor these mutations are non-smokers. The current precision targeted therapy may not be helpful for a large proportion of patients who are smokers [20]. That is because the extensive exposure to carcinogens from smoking frequently results in *TP53* mutation or loss of other tumor suppressor genes, such as *RB1* and *PTEN*, which further induces a high mutational load [20]. These genetic alterations also apply to oral cancers, which occur most frequently in smokers [7]. In the large-scale study of genomic characterization of head neck squamous cell carcinoma (HNSCC) by The Cancer Genome Atlas Network, the major alterations are inactivating mutations of tumor suppressor genes, *TP53* mutation (84%) and *CDKN2A* mutation or deletion (58%) in HPV negative HNSCC cases [7]. Those known druggable fusion oncogenes, such as ALK, ROS or RET genes, reported in other solid tumors, are not observed in HNSCC. So far, the successful druggable targets are inhibitors to overexpressed kinase activity or antibodies binding to the overexpressed receptors [8,12]. *TP53* is the most frequently mutated gene in cancer [21,22] and has been considered undruggable for a long time [23]. Although more detail understanding of the molecular mechanisms of mutant *p53* has progressed and some compounds to restore wild-type-like *p53* function have been developed, the target therapies for *p53* have not reached the clinic to date [21,23,24]. Therefore, future developments to identify novel targets that drive carcinogenesis in smokers should be further implemented.

Similar to the precision medicine applied in NSCLC patients, two major approaches might be taken into consideration. First, broad use of genome-sequencing to profile the genetic landscape of oral cancers should be performed to establish the foundation of precision medicine. This is the use of genetic profiles to identify the selected patients who have the known druggable targets, such as activation of PIK3CA in 34–56% of HNSCC patients and HRAS mutation identified in 5% of HNSCC patients [7]. Second, further identifying and targeting the drivers for oral cancers should be performed. As mentioned, the driver oncogenes show more promising results as targets for developing targeted therapies. We propose to separate head and neck cancers into oral cancer and oropharyngeal cancer groups, separate them again into risk factors-associated or -non-associated groups, then use NGS to identify cancer drivers or screening of known druggable targets such as EGFR, FGFR and PIK3CA alterations, HRAS, CCND1 and MYC mutations, which have been identified in oral cancers [7], in addition to PD-L1 in these patients, to reach the initial precision medicine in oral cancer patients (Figure 4).

## 3. Diagnosis and Treatment

### 3.1. Diagnostic Methods—Biomarkers, Biomarker Test and Definition for Good Biomarkers and Biomarker Tests

Traditionally, tumors are diagnosed and classified based on the histopathological and immunohistochemical characteristics. All tumors use The American Joint Committee on Cancer (AJCC) staging system, which using tumor size and/or depth, lymph node metastasis status and distant metastasis (TNM staging) to predict the prognosis. However, as the classic histopathological diagnosis can only partly predict the prognosis, new molecular tests and methods in addition to morphology-based diagnosis are widely used as a new standard of care in many tumors.

Targeted therapy implies the use of a specific treatment in a selected group of patients whose diseases have the target concerned. Thus, diagnosis based on the validated biomarkers that identify the right patients is the one of the keys for proper targeted therapy. Proper tumor biomarkers need proof of their benefits for improving patient care. A bad biomarker test is as bad as a bad drug [25]. However, determination of a biomarker for the target and establish a test for tumor biomarker are never easy. There are also different intended use contexts for tumor biomarker tests. We will mention below.

The NIH Biomarkers Definition Working Group defined a biomarker as a characteristic that is objectively measured and evaluated as an indicator of normal biologic processes, pathogenic processes, or pharmacological responses to a therapeutic intervention [26]. A biomarker can be used either for evaluation of the prognosis, or for prediction of the outcomes, or both. As a prognostic marker, the presence of a biomarker is associated with better or worse survival compared with the absence of it. As a predictive marker, the presence of a biomarker is associated with a greater difference in treatment outcomes when comparing two treatments or different groups of patients [25]. These tumor biomarkers might be detected or monitored in the blood, tissue or other secretions. The tumor biomarker tests are then used to identify or measure the changes reflected by the tumor biomarkers. For example, the EML4-ALK fusion gene in lung cancers was identified by PCR sequencing initially and the efficacy of ALK inhibitors treatment in these selected patients were using PCR sequencing as the testing method. Later, fluorescence in situ hybridization (FISH) and immunohistochemical (IHC) staining as detection methods for EML4-ALK fusion were established. Treatment outcome based on the FISH or IHC methods needs to be further verified before the new methods can be accepted as biomarker tests. 

### 3.2. Evaluation of the Efficacy of Identification or Detection Methods

To evaluate the efficacy of identification or detection methods, the most important issue is to establish the intended use or context. The common intended use contexts for tumor biomarker tests include risk categorization, screening for cancer, differential diagnosis (such as benign vs. malignant; different lineage of cells; different organs), prognosis (either early stage or metastatic), prediction of therapy activity (either early stage or metastatic) and monitoring disease status (either early stage or metastatic) [27]. As in risk categorization, if the tumor biomarker does not accurately identify the patients at risk and vice versa, then the patients with risk might not get the appropriate treatment or the patients without risk might suffer from the side effects of non-necessary treatment. 

The successful development of precision therapy is based on knowledge of a specific target that drives tumor growth, validation of a clinically applicable biomarker, acceptance of a reasonable end point and understanding of the mechanisms for resistance. Usually, for each specific target, there will be a biomarker developed concurrently. These biomarkers might be either in tissue or blood DNA. There are some essential aspects for evaluating the efficacy of biomarkers, such as analytical validity, clinical validity and clinical utility [28]. For analytical validity, the key question is that does the tumor biomarker test accurately and reliably measure the analyte of interest in the appropriate patient specimen? For clinical validity, the key factor is whether the tumor biomarker tests accurately and reliably identify a clinically or biologically defined disorder, or separate one population into two or more groups with distinct clinical or biological outcomes or differences. For clinical utility, the key factor is whether there are high levels of evidence that use of the tumor biomarker test to guide clinical decisions results in improved measurable clinical outcomes compared with those if the biomarker test results were not applied [25,28,29]. Thus, before considering of a biomarker as a diagnostic identification or detection method, verifying the analytical validity, clinical validity and clinical utility should be carefully performed. 

### 3.3. Development of Biomarkers and Accompanied Treatments in NSCLC

Precision medicine for NSCLC has encountered two major paradigm shifts. The first paradigm shift is the “EGFRoma” first diagnosis policy [30]. Once the tissue samples were diagnosed as adenocarcinoma, then EGFR Tyrosine kinase inhibitor (TKI) was used as the first line therapy. However, since not all patients would have benefited from the therapy, the idea of patient selection before therapy was then started. Among these studies, the clinical and pathologic profiles of the patients who would benefit from EGFR TKI were first defined as Asian, female, never smoker and tumor type as adenocarcinoma [31]. Then, the molecular levels of EGFR, such as protein expression through immunohistochemistry (IHC), gene amplification through fluorescence in situ hybridization (FISH) and gene mutation through PCR, with the clinical outcomes were examined as multiple clinical trials. Therefore, it is a long journey to define the association between EGFR TKI and EGFR mutation tests for lung cancers [8,19,30,32,33,34,35,36,37]. Defining the clinical impacts of a gene mutation is also sometimes a long journey [35].

However, once through the developing journey of EGFR TKI and its molecular tests, more and more gene alterations and their corresponding tests were identified and defined (Figure 3). Then, the problem is the tissue is limited if multiple molecular tests are needed. Furthermore, the hotspots sequencing might miss some uncommon variants or complex variants. Therefore, next generation sequencing (NGS) is required, as it can test multiple genes simultaneously, detect uncommon variants, detect complex variants, define structural and sequence alteration and perform complex biomarker calculation [9,36]. The decision of prescribing target therapies has become more complicated as increasing uncommon variants are detected [9,38,39]. Therefore, some computer-aided software or platform, such as OncoKB or Oncomine, are used to analyze the NGS data and have multiple recommended target therapies illustrated together. Thus, this is the second paradigm shift, from single gene testing to using NGS to identify multiple genes, uncommon variants, and dynamic gene alterations, simultaneously, in NSCLC treatment. 

### 3.4. Current Treatments and Challegenes in Oral Cancer

The treatment of oral cancers is mainly based on NCCN guidelines [15] as in other cancer types. The recommended treatment options in NCCN guidelines are based on the categories of evidence and consensus, and the definitions for each category are listed as category 1: based on high level evidence, there is uniform NCCN consensus that the intervention is appropriate; category 2A: based on lower-level evidence, there is uniform NCCN consensus that the intervention is appropriate; category 2B: based on lower-level evidence, there is NCCN consensus that the intervention is appropriate and category 3: based upon any level of evidence, there is major NCCN disagreement that the intervention is appropriate. Therefore, the clinicians can recommend patients to receive the most appropriate treatments based on the evidence level provided in NCCN guidelines.

The choice of treatment in oral cancer is mainly based on the stage of the disease [40]. For patients presenting with early-stage disease (stage I or II), single-modality treatment with surgery or radiotherapy is generally recommended. Compared to other cancers in the body, oral cancers are easily accessible, and thus surgery is the first choice of treatment and is associated with high cure rates [41]. Combined therapy is generally recommended for patients with locally or regionally advanced disease at diagnosis. The current approved targeted drugs in head and neck cancer include cetuximab and pembrolizumab [5,42,43,44,45,46,47,48]. The timeline of these major therapeutic approaches in recurrent/metastatic head and neck squamous cell carcinoma is illustrated in Figure 5. The efficacy of these targeted drugs will be described below.

Post-operation radiotherapy (RT) or RT plus cisplatin are recommended for patients with high-risk diseases, such as two or more involved nodes, positive margins or extracapsular nodal extension of tumor, presence of perineural invasion or vascular permeation or nodal involvement at levels IV and V based on NCCN guidelines [15]. Adjuvant chemoradiotherapy after surgery is recommended for patients with extranodal extension with or without positive surgical margins. Clinical trials of adjuvant chemoradiotherapy with regimens of cetuximab (a monoclonal antibody directed against EGFR) either combined with docetaxel or weekly cisplatin demonstrated the docetaxel regimen having favorable outcome, with improved disease-free survival (DFS) and overall survival (OS) relative to controls, and has commenced formal testing in a phase II/III trial [49]. For locoregionally advanced disease, concurrent systemic therapy/RT using high-dose cisplatin or carboplatin/5-FU as treatment regimens, which is considered as category 1 preferred option in NCCN guidelines, is recommended. Cetuximab with concurrent RT, carboplatin/paclitaxel and weekly cisplatin, 5-FU/hydroxyurea, cisplatin with infusional 5-FU and cisplatin/paclitaxel are category 2B options [50,51]. For metastatic diseases, cisplatin-based combination regimen (cisplatin/5-FU) has a higher response rates than using single-agent therapy with cisplatin, 5-FU or mexotrexate. Cetuximab plus cisplatin/5-FU or carboplatin/5-FU have improved the response rate and median survival compared to standard chemotherapy of platinum/5-FU [52,53,54,55,56]. 

Similar to lung cancers, EGFR overexpression is common in head and neck squamous cell carcinomas and is associated with poor survival outcomes [57,58]. However, the favorable response to small molecule inhibitors in the case of activating *EGFR* mutations in lung adenocarcinoma has not been observed in squamous cell carcinomas [59]. By contrast, some beneficial effects from the anti-EGFR antibodies are even seen in cases without *EGFR* mutations [60]. Due to the unclear mechanism, no biomarker tests for anti-EGFR treatment are available for oral cancers. Further investigations to identify biomarkers which can be used to identify the patients who will be beneficial for anti-EGFR targeted therapy are required.

Immunotherapy has transformed the treatment landscape of head neck squamous cell carcinoma (HNSCC). Clinical trials evaluating immune checkpoint inhibitors demonstrated efficacy in patients with recurrent or metastatic HNSCC [42,61,62]. The efficacy of pembrolizumab, an anti-PD-1 antibody, was evaluated in patients with a biomarker test, PD-L1 combined positive score (CPS) of ≥20 or ≥1, either as the first-line monotherapy or combined pembrolizumab with chemotherapy and have been observed to have greater median duration of response compared to the EXTREME (Erbitux in First-Line Treatment of Recurrent or Metastatic Head and Neck Cancer) regimen [61]. Thus, immunotherapy as the preferred first-line systemic therapy option for all patients with recurrent, unresectable, or metastatic disease who have no surgical or radiotherapeutic option is now suggested by NCCN guidelines. 

The efficacy of Nivolumab was assessed in a clinical trial including patients with recurrent HNSCC whose disease had progressed after platinum-based chemotherapy [62]. Significantly greater overall survival, one-year survival and response rate in patients treated with nivolumab compared to patients with a standard second line single agent systemic therapy (methotrexate, docetaxel, or cetuximab) was observed. Better overall survival was confined to patients with a tumor PD-L1 expression level ≥1%. 

Due to these promising clinical trials, immunotherapy (nivolumab and pembrolizumab) is now considered as an NCCN category 1 preferred option for patients with recurrent or metastatic HNSCC who have progressed on from, or following, platinum-based chemotherapy [63]. Even though there are still some ambiguities about the PD-L1 testing and definitions, PD-L1 expression may be associated with better outcomes from treatment with immunotherapy for recurrent or metastatic HNSCC [15]. Thus, a PD-L1 test is the only currently available biomarker test for oral cancers. 

So far there is no biomarker which can achieve clinical validity and clinical utility, except PD-L1 testing, in oral cancers. However, instead of analyzing a single gene or single or multiple markers, which was the first paradigm shift in precision medicine in NSCLC, using NGS or even multi-omics data, which has become a common tool to analyze broad genetic and molecular alterations, to correlate with treatment outcomes might be a new paradigm shift for precision medicine in oral cancers [7,64,65,66,67,68]. 

## 4. Monitor

### 4.1. Monitoring Methods

Although molecular-targeted therapies can be effective initially, considerable fractions of patients’ treated tumors will eventually become resistant and progress. Understanding the mechanisms of resistance will help to manage patients’ disease progression and may prevent or disrupt the process. Furthermore, patients with less tumor volume show better prognosis. Thus, how to monitor the disease progress is also an important issue in precision medicine.

The current standard for monitoring tumor progression is through rigorous use of diagnostic imaging. However, long-term frequent imaging could be a laborious process. Moreover, even with the improvement of sensitivity of imaging, tumors cannot be detected until they reach certain size. A new tissue specimen for proving diagnosis, evaluating new genetic changes and understanding the mechanisms of resistance is also recommended. In addition to standard clinicopathological evaluation including imaging studies, molecular characterization using NGS and/or other omics-based tests is an emerging way to monitor disease progress [69,70]. In the development towards precision medicine, there are some newly developed methods beneficial for monitoring tumor progression. Selected new methods will be briefly mentioned below.

#### 4.1.1. Patient-Derived Tumor Xenograft (PDX) Models

Murine models transplanted with cancer cell lines or human tumor cells have been commonly used tools to study a variety of biological events in biomedical research, including tumor initiation, progression and responses to treatments. Usually, these human tumor models are generated in immunodeficient mice by subcutaneous or orthotopic injection of tumor cell lines that have been propagated in culture for many passages. Due to long term manipulation of the tumor cells, this system is quite different from the original tumors. Thus, these conventional xenograft models have limited benefits for further drug screenings or predict the pre-clinical efficacy of treatments [71]. To recapitulate patients’ responses to therapy depends on models accurately reflecting patients’ specific molecular events. Hence, patient-derived tumor xenografts (PDXs) have been established as a preclinical platform for assessment of drug efficacy [72]. Instead of injection of tumor cells as single cells, PDXs have been established by implantation of small pieces of tumor tissues derived from cancer patients into highly immunodeficient mice. PDX models maintain the histopathological structures, genomic and gene expression profiles of the original tumors [71,73,74]. Accumulating evidence suggests the effectiveness of PDX models in predicting the efficacy of anti-cancer therapies, which can be applied in precision medicine. In order to avoid human stroma replaced by murine stroma, mouse anti-human response and feasibility of evaluating human immune system in the murine model, the next generation of murine humanized models with humanized immunity have been developed through engrafting human immune cells from either bone marrow, liver, thymus (BLT) or CD34+/hematopoietic stem cells (HSCs) or human peripheral blood mononuclear cells (PBMCs). These murine humanized models with humanized immunity can be further used for investigating clinical applications of cancer immunotherapeutic agents [75].

#### 4.1.2. Tumor Organoids

Recently, organoids, the three-dimensional (3D) constructs which are comprised of multiple cell types originating from stem cells and are capable of simulating the function and architecture of original tissues or organs, have been developed and widely used in vitro and in vivo studies [76,77]. Tumor organoids have been used and been proven effective as preclinical models to predict personalized response to therapies [77,78]. The tumor organoids have the potential to accurately recapitulate the intra- and intertumoral biological heterogeneity associated with patient-specific tumors. Thus, if the reproducible platforms for culture and propagate tumor organoids can be established, this system could accelerate translatable insights into patient care. Compared to PDX, organoid culture of human tumor tissue has emerged as a relatively low-cost and representative platform to model cancer heterogeneity and interactions with the tumor microenvironment in vitro [79].

#### 4.1.3. Liquid Biopsy

Most of the common cancers, such as lung cancers, breast cancers and colorectal cancers, are derived from internal organs. Biopsies to get a tissue specimen for further evaluation are invasive and unpleasant procedures for patients. Through the advances in technologies, detection of blood-based, tumor-specific biomarkers, such as circulating tumor cells (CTCs) or circulating cell-free tumor DNA (ctDNA), is now feasible and showing some convincing results [8,80,81,82]. The recently developed high-sensitivity liquid biopsy assays have enabled the identification of minimal residual disease in patients. Emerging evidence has shown the predictive value of CTC detection at primary diagnosis of cancer and its potential in making assessments for monitoring post-surgical relapse [82]. 

### 4.2. Current Recommended Guideline for Moitoring Recurrence in Oral Cancers and Challenges

For patients with locoregionally advanced disease who have undergone surgery, post-operative imaging is recommended by NCCN guidelines if there are signs of early recurrence or for patients considered at high risk of early recurrence. After definitive-intent treatment completion, imaging 3–4 months after the end of treatment is recommended. If there is a concern about an incomplete treatment response, then imaging can be performed as early as 4–8 weeks after treatment. The image study choices include computed tomography (CT) and/or magnetic resonance imaging (MRI) with contrast and fluorodeoxyglucose positron emission tomography (FDG PET)/CT. PET/CT is preferable imaging, which shows 52.7% and 96.3% positive predict value (PPV) and negative predict value (NPV) for detecting local residual or recurrent disease and 72.3% and 88.3% PPV and NPV for detection of nodal residual or recurrent disease [15,83]. If PET/CT is used for follow-up, the first scan should be performed at a minimum of 12 weeks after treatment to reduce the false-positive rate [83,84,85]. 

Unlike the malignancies in the internal organs, oral precancerous or cancerous lesions can be identified using the naked eye. Upon careful examination, experienced specialists, such as oral pathologists or oral surgeons, can recognize lesions without much difficulty. Local recurrence or regional relapse are the major issues for oral cancers. Due to field cancerization, new tumors can grow from positive or dysplastic margins or even other previous dysplastic regions. Unfortunately, although we can recognize the lesions, no preventive treatment to reverse or cease the tumorigenesis process is available. Eradicating the precancerous lesions by either blade or laser or cryotherapy or photodynamic therapy is the current treatment recommendation. Due to field cancerization, nearly “normal” section margins can not be achieved in patients with risk factors. The other major challenges of monitoring oral cancers or precancerous lesions are encountered when patients have severe trismus either due to submucosal fibrosis caused by a betel nut chewing habit or when tumor nests are growing underneath the dense fibrotic scaring surface. The potential of liquid biopsy in head and neck cancer has been studied and might have potential application for monitoring these patients in the future [86]. The aforementioned PDX and tumor organoids might also help to develop and evaluate novel treatments for the oral cancer patients in the future (Figure 6).

### 4.3. Future Directions

From a review of the information of clinical presentations, imaging studies and pathological features, molecular characterization using NGS and/or other omics-based tests, are emerging ways to diagnose and monitor the disease progress in precision medicine. Due to the large number of data that will be acquired and processed, utilizing machine learning and artificial intelligence to reach deep phenotyping which links the clinical abnormalities and molecular states will facilitate progress in precision medicine.

## 5. Conclusions

New molecular tests and methods, in addition to morphology-based diagnosis, are widely used as a new standard of care in many tumors. “One-size-fits-all” medicine is now shifting to precision medicine. We have reviewed the journey toward to precision medicine in leading cancers, especially NSCLC, and the implication and challenges of precision medicine in oral squamous cell carcinoma.

## Figures and Tables

**Figure 1 jpm-12-00012-f001:**
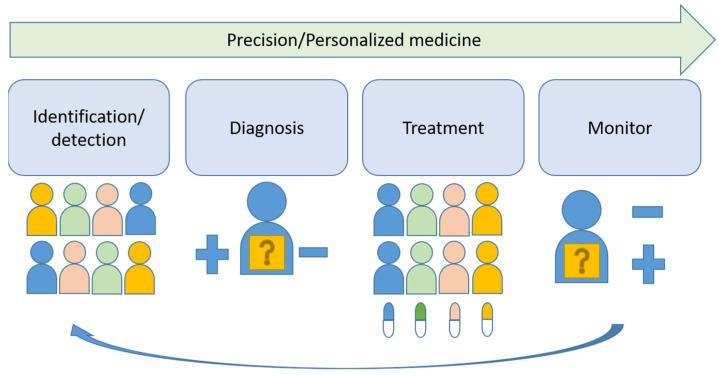
Schematic illustration of key steps of precision medicine.

**Figure 2 jpm-12-00012-f002:**
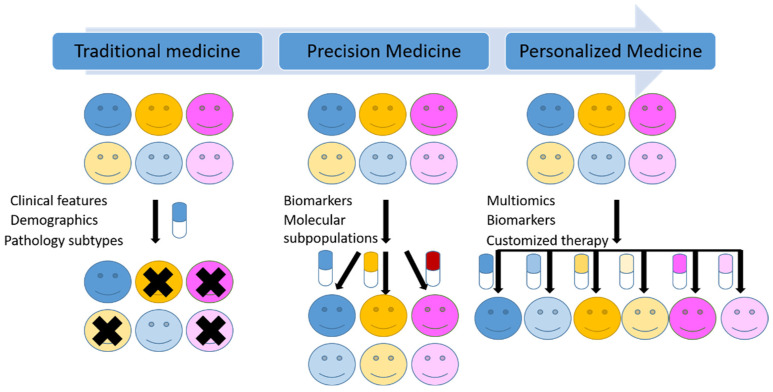
Schematic illustration of the transition from traditional medicine to precision medicine and future personalized medicine.

**Figure 3 jpm-12-00012-f003:**
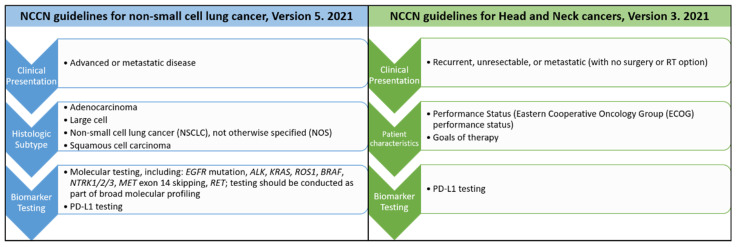
Comparison of the most recent treatment guidelines from The National Comprehensive Cancer Network^®^ (NCCN^®^) for non-small cell lung cancers and head and neck cancers.

**Figure 4 jpm-12-00012-f004:**
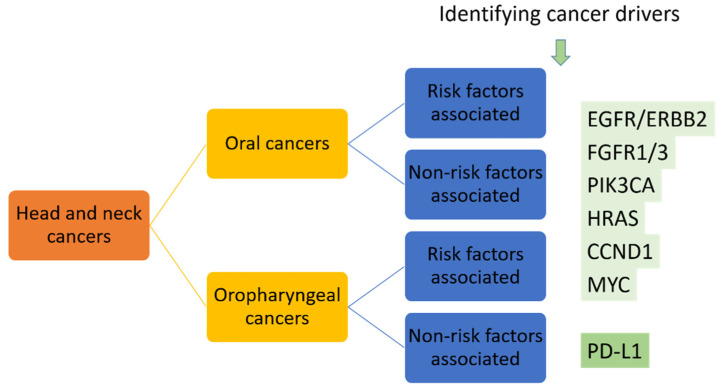
Proposed targeted therapy for head and neck cancers based on the identified druggable targets.

**Figure 5 jpm-12-00012-f005:**
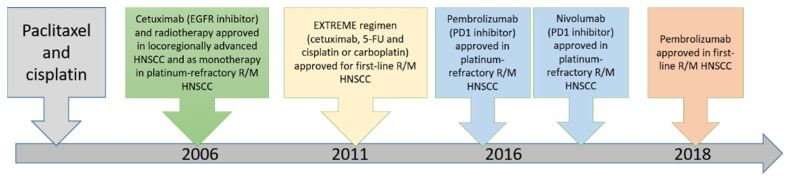
Timeline of major discoveries and related therapeutic approaches in head and neck squamous cell carcinoma. (R/M: recurrent/metastatic, HNSCC: head neck squamous cell carcinoma).

**Figure 6 jpm-12-00012-f006:**
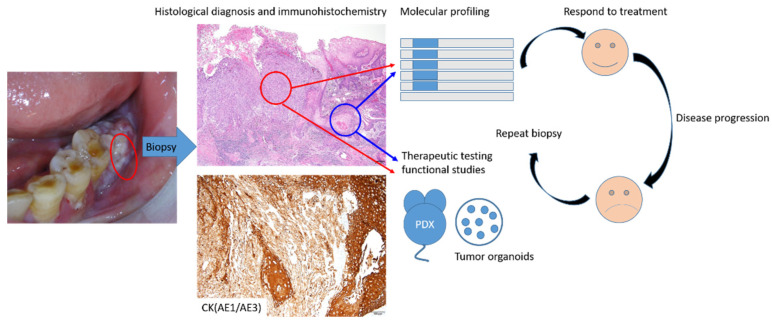
Schematic illustration of precision medicine in oral cancer in the future. PDX, Patient-Derived Tumor Xenograft.

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
