# Peer review of "Contemporary Molecular Analyses of Malignant Tumors for Precision Treatment and the Implication in Oral Squamous Cell Carcinoma"

_jpm, 2021, doi:10.3390/jpm12010012_

Round 1
Reviewer 1 Report
The manuscript entitled ”Contemporary molecular analyses of malignant tumours for precision treatment and the implication in oral squamous cell carcinoma” aims to be a review that focuses on various treatment options for several malignant disorders and their connection with OSCC. The structure of the entire review article is confusing, it does not point out separately the type of malignant tumours, in one sentence is one affirmation and in the other the idea changes and debates other type of malignant tumour. I consider that the title is not at all appropriate and related the idea of the review, the introduction should as well provide information related to the molecular diagnosis and options in case of OSCC and the wide range of potentially biomarkers, as the entire information focuses on other types of tumour examples. For example, in the identification of the target paragraph - there is no related information to OSCC. A statistical analysis and comparison of the included analysed research articles is missing. In order to compare, more results need to be analysed and their conclusions stated. For example, there is a focus on the TP53 gene alteration and its link to lung and oral cancer, but the authors should take into consideration as well the results of the studies with different outcomes. In the first part of the study, the focuses seems to be on lung cancer and the existent information related to OSCC is discussed from a general vague perspective, not being supported by existent relevant studies. This manuscript does not have the discussion section, section that in case of reviews should be also included. Nevertheless, this review does not follow the PRISMA guidelines.
Author Response
Response:
- We thank the editors and the reviewers for their thoughtful evaluation of our manuscript. To make the structure of this article clearer, we have added a schematic drawing to illustrate the structure of this article.
- This manuscript type is a perspective and aimed to introduce the key steps toward to precision medicine using NSCLC as main example. Since this manuscript is not conventional systemic reviews or meta-analyses, thus the Preferred Reporting Items for Systematic Reviews and Meta-Analyses (PRISMA) guideline can not be applied. However, the major conceptual framework is based on world-wide well accepted guideline posted by The National Comprehensive Cancer Network (NCCN). Majority of the articles related to the current oral cancer treatment in our manuscript are also cited by NCCN official articles.
- Because the non-small cell lung cancer (NSCLC) is one of the most advanced cancer type utilizing the principle of precision medicine, we mainly used NSCLC as the role model to demonstrate the 20-year process how the treatment of NSCLC toward to the current status. Unfortunately, we can see these kind of studies in oral cancers are much behind. This is the reason we wrote this perspective and would like to provide a broader view how precision medicine can be applied in oral cancers.
- We agreed with the reviewer’s point view. There are too many “targets” be stated in oral cancers. A statistical analysis, comparison of the included analyzed research articles based on PRISMA guidelines should be used for a review focusing on these potential targets. However, this is not the main theme of our manuscript, thus we did not particularly mention the unclear targets in this manuscript to avoid bias.

Reviewer 2 Report
This is a well-written paper in which the authors review the key steps toward the progress of precision medicine and the implication in OSCC.
LINE 280: Please change to the same font as other text
Author Response
Response: We have corrected Line 280 accordingly. We thank the editors and the reviewers for their thoughtful evaluation of our manuscript.

Reviewer 3 Report
This paper concerns a comprehensive review on new targeted testing for oral squamous cell carcinoma. As this still is a major topic in the literature, with new advancements coming out from primary reaserch, a thorough understanding of the basis of research methodology is required.
This review does contain all the basis for the current non-surgical treatment options for oral cancer, with a direct comparison with lung cancer. The manuscript presents all main concepts in a clear way, with no major setbacks on content. In addition, future concepts are clearly visualized and expressed with no leaps in knowledge. The literature review is considered adequate and the overall use of English is more than acceptable.
All in all, I strongly recommend the publication of this manuscript to the Journal of Personalized Medicine, as it can surface the major issued associated with the medical treatment on head and neck malignancies.
Author Response
Response: We thank the editors and the reviewers for their thoughtful evaluation of our manuscript.

Round 2
Reviewer 1 Report
I appreciate the modifications and if the editor and the other reviewers consider that the article is suitable for your journal it means that it can be published.